

# Deep learning based cardiac disorder classification and user authentication for smart healthcare system using ECG signals

Tong Ding[1], Chenhe Liu[2], Jiasheng Zhang[3], Yibo Zhang[4,5] and Cheng Ding[6]

[1] School of Engineering, University of New South Wales, Sydney, Australia
[2] Yew Chang International School Beijing, Beijing, China
[3] Department of Electronic Commerce, Anhui International Studies University, Wuhu, Anhui, China
[4] Gezhi Future Research Institute, Beijing, China
[5] School of Systems and Computing, University of New South Wales, Sydney, Australia
[6] Department of Biomedical Engineering, Georgia Institute of Technology, Atlanta, GA, United States

## ABSTRACT

Abnormal cardiac activity can lead to severe health complications, emphasizing the importance of timely diagnosis. It is essential to save lives if diseases are diagnosed in a reasonable timeframe. The intelligent telehealth system has the potential to transform the healthcare industry by continuously monitoring cardiac diseases remotely and non-invasively. A cloud-based telehealth system utilizing an Internet of Things (IoT)-enabled electrocardiogram (ECG) monitor gathers and analyzes ECG signals to predict cardiac complications and notify physicians in crises, facilitating prompt and precise diagnosis of cardiovascular disorders. Abnormal cardiac activity can lead to severe health complications, making early detection crucial for effective treatment. This study provides an efficient method based on deep learning convolutional neural network (CNN) and long short-term memory (LSTM) approaches to categorize and detect cardiovascular problems utilizing ECG data to increase classifications (referring to distinguishing between different ECG signal categories) and precision. Additionally, a threshold-based classifier is developed for the telehealth system's security and privacy to enable user identification (for selecting the correct user from a group) using ECG data. A data preprocessing and augmentation technique was applied to improve the data quality and quantity. The proposed LSTM model attained 99.5% accuracy in the classification of cardiac diseases and 98.6% accuracy in user authentication utilizing ECG signals. These results exhibit enhanced performance compared to conventional machine learning and convolutional neural network models.

Corresponding author
Cheng Ding, chengding@gatech.edu

## INTRODUCTION

The Worldwide Health Organization (WHO) claims that heart disease is the leading cause of mortality worldwide. Approximately 15 million deaths worldwide are attributed to heart disease or 31% of all fatalities (*Luo et al., 2024*). Heart attacks and strokes account for four

out of every five cardiovascular fatalities, and one-half of these deaths happen before age 60. Most birth defect-related fatalities are caused by heart abnormalities, among the most prevalent congenital disabilities. About one in 125 newborns are born each year with congenital cardiac abnormalities. The flaw might be so little that the infant seems normal for several years after birth, or it could be so severe that it poses an immediate threat to the individual's life (*Heron, 2019*). Electrocardiography records cardiac electrical impulses, delivering more comprehensive insights than conventional sonographic techniques, as numerous cardiovascular conditions display unique structural patterns. An intelligent healthcare system does not experience weariness; therefore, it can analyze large datasets at a significantly higher velocity and with greater precision than people (*Hassaballah et al., 2023*). The electrocardiogram (ECG) is a useful non-invasive instrument for various biological tasks, including heart rate measurement, cardiac rhythm analysis, heart abnormality diagnosis, sentiment analysis, and biometric authentication (*National Center for Health Statistics (US), 2007*).

Artificial intelligence (AI) is used in various medical fields, including cancer, radiography, dermatology, neuroscience, and cardiology (*Shah et al., 2022*; *Acharya et al., 2017*). Recent breakthroughs in the Internet of Things (IoT), wearables, and sensing technologies have enhanced the quality of healthcare services, resulting in a transition from traditional clinical-based healthcare to real-time monitoring (*Khanna et al., 2023*). The current approaches are increasingly investing in merging various technologies like machine learning and deep learning (DL) algorithms with IoTs, Cloud services, and big data to provide advanced e-Health mechanisms that exploit the growth of online connections for a medical condition. DL is a machine learning methodology that employs multi-layered networks, wherein first layers extract information and subsequent layers analyze and classify patterns (*Arel, Rose & Karnowski, 2010*). Convolutional neural networks (CNNs), recurrent neural networks (RNNs), and deep belief networks (DBNs) are examples of deep discriminatory models (*Yıldırım et al., 2018*). The DL network is fundamental to detecting heart disorders using frames of ECG signals. A DL method is based on mathematical formulae and functions similarly to the human brain. The mathematical functional principles aim to comprehend and identify patterns among many elements by training itself on the input and output data to establish the patterns among them. The system can identify the entities it has been educated to recognize (*Pławiak, 2018*). *Ardeti et al. (2023)* conducted a review of ECG-based cardiovascular disease monitoring systems utilizing both traditional and deep learning methodologies, underscoring IoT-enabled real-time, cost-effective monitoring solutions. Their findings accentuate enhanced accuracy in early diagnosis and the efficacy of smart healthcare technologies for remote patient management.

Recent developments in ECG signal classification for cardiac disease detection have incorporated diverse deep learning techniques, tackling issues including imbalanced data and improving model efficacy. *Rath et al. (2021)* made a significant contribution by proposing an ensemble model that integrates a generative adversarial network (GAN) with long short-term memory (LSTM) to address the imbalance in ECG data. Their model attained an exceptional accuracy of 0.994 and an AUC of 0.995 on the PTB-ECG dataset,

markedly surpassing other models, including naïve Bayes. *Śmigiel, Pałczyński & Ledziński (2021)* conducted research on ECG signal classification utilizing the PTB-XL dataset and revealed that a Convolutional Neural Network (CNN) incorporating entropy-based features achieved the highest classification accuracy. They emphasized that a conventional CNN model provided superior computing efficiency. In an alternative setting, *Loh et al. (2023)* devised a 1D CNN model to categorize patients into ADHD, ADHD+CD, and CD, attaining an exceptional classification accuracy of 96.04%. They augmented the model's explainability with Grad-CAM, which offered temporal insights to facilitate clinical interpretation. Additionally, *Abubaker & Babayiğit (2023)* introduced a CNN-based model aimed at classifying four primary cardiac anomalies utilizing ECG image data, attaining an impressive accuracy of 98.23%. Their methodology, when integrated with naïve Bayes for feature extraction, achieved a maximum performance of 99.79%, exceeding that of pretrained models like SqueezeNet and AlexNet. Furthermore, *Aarthy & Mazher Iqbal (2024)* presented a deep learning methodology incorporating fine-tuned CNNs and clustering algorithms, designed to identify minor discrepancies in ECG readings. Their methodology demonstrated enhanced efficacy relative to current methodologies, facilitating the early identification of cardiovascular illnesses by the detection of subtle anomalies in ECG data. Biometrics is now universally acknowledged as more trustworthy than knowledge and skills and possession-based systems such as identification cards and login information since there is no need to memorize anything with biometrics. Due to their extreme difficulty in forging and the requirement that a real user is present to allow access to specific resources, biometric traits provide better security. They cannot be misplaced, exchanged, or pirated. Consequently, many healthcare systems incorporate biometrics based on ECG to enable continuous authentication (*Chawla, 2020*; *George & Ravindran, 2020*). Figure 1 shows the intelligent healthcare system schematic diagram.

Advancements have been achieved by integrating several signal kinds and implementing more intricate architectures to enhance classification accuracy and robustness. *Hangaragi et al. (2025)* created a multi-class classification model that combines ECG and phonocardiogram (PCG) signals for thorough heart disease identification. Their model attained an accuracy of 97% with a minimal error rate of 0.03, illustrating its efficacy in categorizing six unique cardiac diseases. *Karapinar Senturk (2023)* suggested a deep learning-based approach for ECG classification that employed scalogram pictures produced *via* continuous wavelet transform (CWT). Their AlexNet-based model attained an accuracy of 98.7%, markedly surpassing a self-developed CNN. This work highlighted the efficacy of preprocessing approaches, such as CWT, in enhancing model performance for the categorization of arrhythmia, congestive heart failure, and normal sinus rhythm. *Khanna et al. (2023)* made a significant contribution by developing an IoT-enabled deep learning model (IoTDL-HDD) for the real-time diagnosis of cardiovascular diseases. This model incorporated a bidirectional long short-term memory (BiLSTM) network for feature extraction, enhanced by the artificial flora optimization (AFO) method, with a fuzzy deep neural network (FDNN) classifier. Their methodology attained a maximum accuracy of 93.45%, illustrating the possibility of integrating IoT with deep learning for real-time clinical applications. These research collectively underscore the many and new

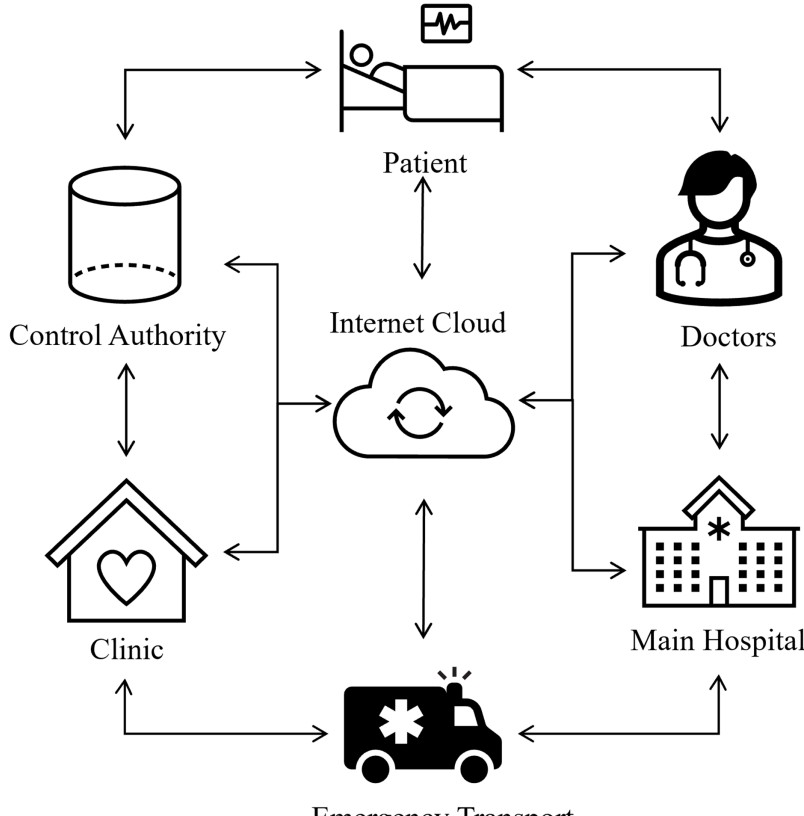

**Figure 1 Smart healthcare system: an intelligent framework for real-time health monitoring and decision support.**

methodologies being investigated to improve ECG signal categorization for heart disease detection. They demonstrate the amalgamation of sophisticated deep learning frameworks, hybrid models, and innovative preprocessing methodologies, all enhancing diagnostic precision and the capability for real-time, automated clinical decision assistance.

Despite the considerable progress in ECG signal classification for cardiac disease diagnosis, some restrictions persist. Numerous models depend on extensive, accurately labeled datasets, which are frequently limited, and their generalizability across diverse datasets or patient populations may be constrained. Moreover, the intricacy of models such as BiLSTM and FDNN can lead to substantial computing requirements, rendering them less appropriate for real-time applications. Integrating various signal types, as demonstrated in certain research, might enhance detection but may result in overfitting and diminished interpretability. Furthermore, the absence of thorough testing in many real-world clinical environments and the difficulty of model interpretability impede wider use. Resolving these difficulties is essential for effective implementation in healthcare.

The approach for classifying and diagnosing ECG information and improving the capacity to identify abnormal cardiovascular activity using deep neural networks is proposed in this study. CNN and LSTM-based DL architectures are designed to efficiently classify the abnormality of cardiovascular activity using ECG signals in a telehealth system.

Additionally, an ECG-based biometric system is programmed for privacy and security for an intelligent health system. The proposed ECG-based user authentication system will help secure the patients' sensitive data. The main contributions of the research study are as follows:

- We design an efficient deep learning-based CNN and LSTM algorithm for classifying cardiac vascular abnormalities in ECG signals.
- We demonstrated the implementation of ECG-based user authentication to refer to verifying a user's identity for telehealth systems using the convolutional and long short-term memory architectures for identification.
- A data preprocessing and data augmentation approach is used to enhance the training data and broaden the variety of the input data to get better results on standard datasets.
- We showed that our algorithms could be deployed to ECG data gathered under various conditions, providing higher or similar consistency to the highest state-of-the-art approaches.

The rest is organized as follows: "Materials and Methods" shows the materials and methods we used to analyze the most popular options. "Result Analysis" investigates the results and discussion of deep learning neural networks in the domain of ECG-based analysis. "Computing Environment" outlines the conclusion and future work.

## MATERIALS AND METHODS

The recommended method with algorithms is thoroughly examined in this section. The literature review that was mentioned before provided the basis for the technique that is presented in this section. To answer each research topic, DL methods were used. To achieve the highest accuracy, the parameters and the number of layers of each DL model have been changed. The processing, training, and evaluation stages of ECG identification are shown visually in Fig. 2. The recommended method is based on algorithms for deep learning that have been trained and tuned using a variety of hyperparameters. Neural networks adapt their convergence speed and biases accordingly.

### Convolution neural networks

Deep learning, a recent innovation in artificial intelligence, demonstrates near-human proficiency in many applications such as autonomous medical diagnosis, speech-to-text translation, feature detection, and classification. The CNN is a type of DL algorithm that can take in information, add biases and weights to different feedback features, and then discriminate among them (*Albawi, Mohammed & Al-Zawi, 2017*). A traditional CNN utilized two-dimensional input, including pictures and videos. As a result, it became commonly known as two-dimensional CNN. A more recent development is the 1-dimensional CNN, a two-dimensional CNN revision. The ECG records supplied the CNN model utilized during extracting features. Six layers make up this structure and alternate among 1-dimensional convolution and 1D max-pooling stages. A sub-CNN algorithm was developed for several sessions based on the latest findings. Enhancing the

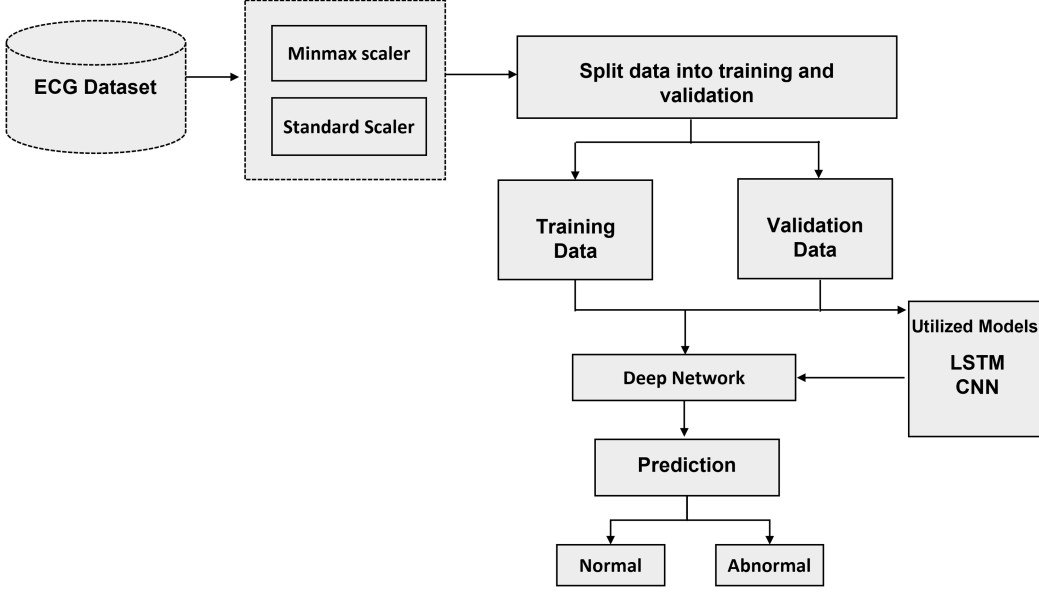

**Figure 2 Schematic representation of the proposed methodology: a step-by-step workflow for intelligent ECG-based cardiac disorder classification and user authentication.**

system depth and the specifications of every level in this procedure may enhance the detection accuracy, whereas the total effect is enhanced.

A feature extraction method may incorporate down sampling and convolutional layers. A kind of fuzzy filter known as convolution can enhance information quality while reducing distortion. The top element's extracted features are convolutional with the current gradient convolution kernel (*Mendieta et al., 2019*). An equation then returns the outcomes of the convolutional calculations. Every generated vector provided as a target is compared against the inaccuracy using the given equations. If it is fulfilled, the learning is finished, and the numbers and limits are stored in the interim. The classifiers are created under the assumption that each weight is stable. Figure 3 shows the schematic representation of the proposed CNN network.

This CNN is precisely developed and methodically tuned through extensive testing to attain optimal performance in data classification. The model architecture comprises three 1D convolutional layers, each succeeded by batch normalization and max-pooling layers, to extract hierarchical features from the input data. Subsequent to these layers, there exists a flattening layer, followed by two completely connected dense layers using dropout regularization, culminating in a softmax layer for multi-class classification. The architecture comprises an input layer, succeeded by a Conv1D layer featuring 64 filters and a kernel size of 6, employing ReLU activation, batch normalization, and max pooling with a stride of (3, 2). This block is reiterated two additional times. The output features are further flattened and processed through two thick layers, each including 64 units with ReLU activation, accompanied by dropout layers at a rate of 0.2 to mitigate overfitting. The final dense layer contains five units and utilizes a softmax activation function for
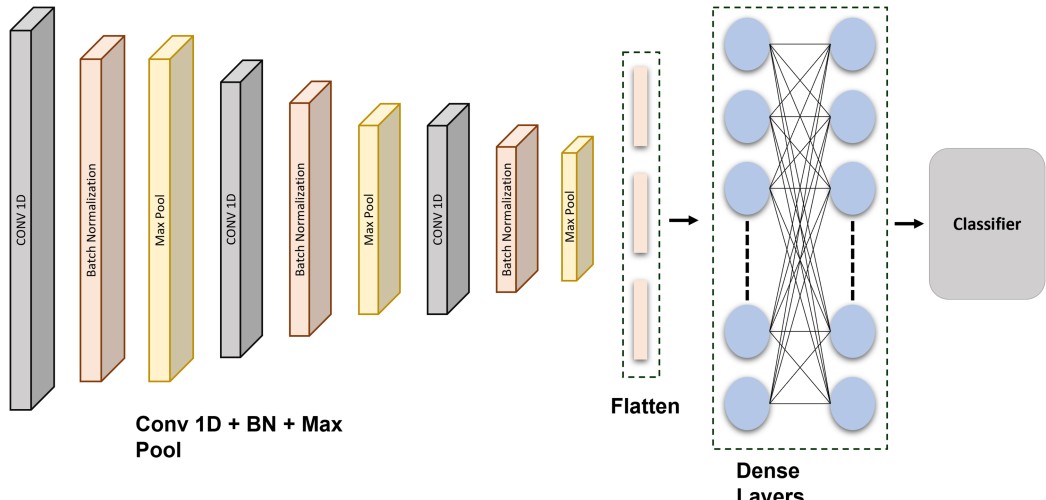

**Figure 3** Schematic representation of the proposed CNN architecture: a layer-wise overview for ECG signal classification and authentication.

classification. Each of the five output units corresponds to a specific class in the dataset, reflecting separate categories that the model is designed to differentiate. The softmax function transforms raw output scores into probabilities, with each unit's output representing the probability that the input sample is associated with the corresponding class. The class with the greatest likelihood is chosen as the definitive forecast. The model employs the Adam optimizer in conjunction with the categorical cross-entropy loss function, facilitating steady and efficient learning.

A grid search approach is used for hyperparameter optimization to enhance the model's performance. This systematic method examines several values for essential hyperparameters, including the quantity of convolutional filters ([32, 64, 128]), kernel dimensions ([3, 5, 6]), and dropout rates ([0.2, 0.3, 0.5]). Every combination is assessed by 5-fold cross-validation. The efficacy of each model configuration is evaluated using validation accuracy and categorical cross-entropy loss, aiming to identify the combination that attains the highest validation accuracy and the lowest validation loss. Of the evaluated configurations, the model using 64 filters, a kernel size of 6, and a dropout rate of 0.2 demonstrates optimal performance. This configuration has been chosen for the final implementation. The tuning technique greatly enhances the model's generalization and robustness. Additionally, early halting is employed during training, contingent upon validation loss, to prevent overfitting. Although learning rate scheduling is examined, the default learning rate of the Adam optimizer is adequate for consistent convergence. This experimental process of parameter tuning and layer tweaking is essential for enhancing the CNN model's classification accuracy.

The CNN was configured with the following layer: Inputs → Convolution-1D (64 filters, map size 6, Relu) → Batch Normalization → Max Pool Layer (stride 3, 2) → Convolution-1D (64 filters, map size 6, Relu) → Batch Normalization → Max Pool Layer (stride 3, 2) → Convolution-1D (64 filters, map size 6, Relu) → Batch Normalization →

Max Pool Layer (stride 3, 2) Flatten Layer → Dense (64 units, Relu) → Dropout (0.2) → Dense (64 units, Relu) → Dropout (0.2) → Dense (5 units, Relu) → SoftMax. The Adam optimization technique with categorical cross entropy produces better outcomes. The output of the convolution layer:

$$X_J^I = f\left(\sum_{i \in M_j} X_J^{I-1} * W_{ij}^I + b_J^I\right). \tag{1}$$

The information shared between hidden layers is represented by:

$$X_J^I = f\left(B_j^I \text{down}\left(X_j^{I-1}\right) + b_j^I\right) \tag{2}$$

the output are calculated by decreasing the error by the following equation:

$$E = \frac{1}{2}\sum_{k=0}^{n-1}(d_k - y_k)^2 \leq \varepsilon. \tag{3}$$

## Long short-term memory

Long short-term memory is a recurrent neural network that can comprehend and predict consecutive information. RNN does have a limited ability to sustain long-term memory. The LSTM was created; as a result to address this problem by incorporating gating operations and a storage cell that can maintain its configuration over the duration (*Sainath et al., 2015*). Figure 4 shows the proposed LSTM model. A memory location, a recall gateway, an initialization vector, and an update gate are the essential parts of the LSTM. The activated sigmoid function manages the input gate's output vector. We provide an enhanced LSTM that is specially made for identifying ECG signals to help solve the issue. An exponent is incorporated into the LSTM in this method to facilitate efficient data distribution and enhance interpretation. The formulas for the current block, potential cell state, and LSTM output are given:

$$\tilde{c}_t = \tanh(w_c[h_{t-1}, x_t] + b_c) \tag{4}$$
$$c_t = f_t * c_{t-1} + i_t * \tilde{c}_t \tag{5}$$
$$h_t = O_t * \tanh(c^t). \tag{6}$$

Equations (4) to (5) delineate the internal mechanisms of a LSTM unit, which is extensively employed for simulating sequential data. In Eq. (4), the prospective cell state. The calculation involves applying a tanh activation function to a linear transformation of the concatenated prior hidden state $h_{t-1}$ and the current input $x_t$. This candidate cell state serves as a prospective enhancement to the memory. Equation (5) modifies the current cell state $c_t$ by integrating the prior cell state $c_{t-1}$, adjusted by the forget gate $f_t$, and the candidate cell state $\tilde{c}_t$. t, modulated by the input gate. This mechanism enables the LSTM to selectively preserve pertinent information from the past and assimilate fresh input. Ultimately, Eq. (6) establishes the current hidden state, which functions as the output of the LSTM unit at time step t. The concealed state is derived by applying the output gate $O_t$ to the tanh activation of the present cell state. The gated mechanisms of the

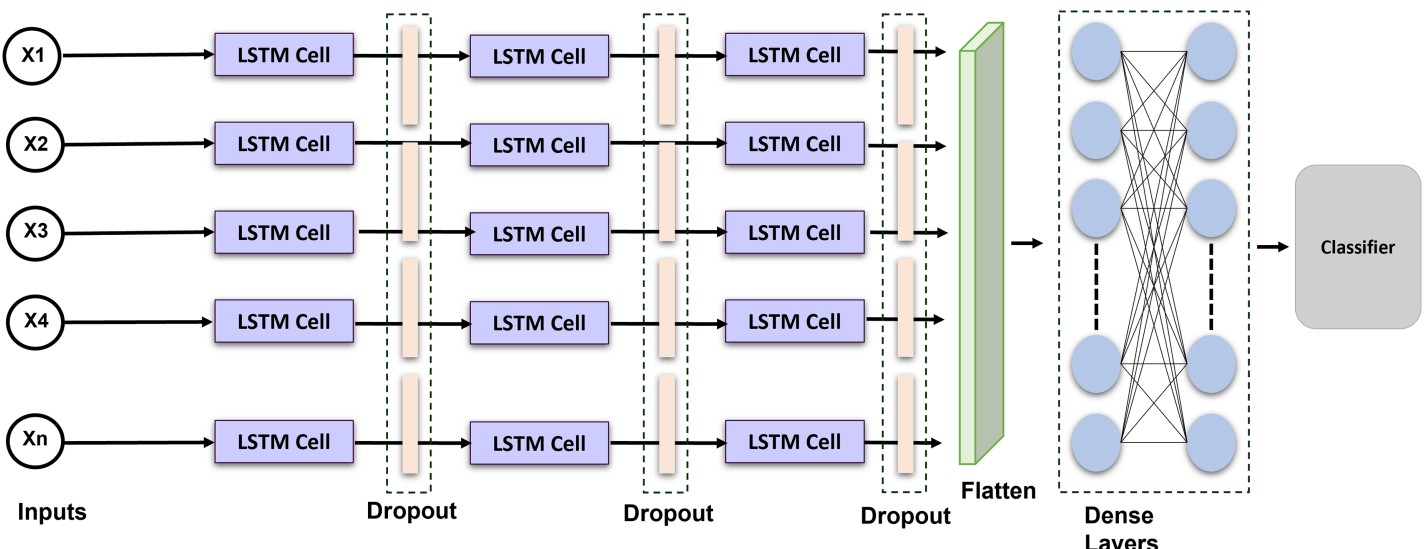

**Figure 4** Schematic representation of the proposed LSTM architecture: temporal modeling for ECG-based cardiac classification and user authentication.                                              

LSTM facilitate the management of long-range dependencies and temporal dynamics, rendering it particularly adept for tasks like ECG signal classification and authentication.

In order to see if the outcomes were notably superior to those of other networks, we introduced a long short-term memory layer to the most current optimization technique. The network's layer was configured as follows: LSTM Layer (16 filters) →, Dropout (0.2 rates) → LSTM Layer (32 filters) → Dropout Layer (0.2 rates) → LSTM Layer (64 filters) → Dropout Layer (0.1 rates) → Flatten Layer → Dense Layers. The selection of 64 filters and a kernel size of 6 was determined through empirical testing and literature demonstrating their efficacy in detecting pertinent temporal patterns in ECG signals. Batch normalization was employed to stabilize and expedite training, whilst max pooling facilitated dimensionality reduction and computational efficiency. Dropout layers with a rate of 0.2 were utilized to mitigate overfitting. The model was optimized with Adam and categorical cross-entropy, resulting in improved convergence and classification accuracy. These decisions together enhanced performance in multi-class ECG classification tasks. The Adam optimization technique with categorical cross-entropy yields superior outcomes.

The framework incorporates LSTM layers because of their proficiency in capturing long-range relationships in sequential data. LSTMs are adept at handling tasks with temporal dynamics due to their ability to save knowledge throughout prolonged time intervals through internal memory cells and gating processes. The architecture employs layered LSTM layers to augment the model's ability to learn intricate temporal characteristics. This study selects LSTM networks above gated recurrent units (GRUs), despite the latter's computational efficiency and reduced parameters, due to LSTMs' superior performance on datasets characterized by extended and complex temporal dependencies. LSTMs provide enhanced control over information flow *via* distinct input,

output, and forget gates, which is beneficial for modeling intricate temporal patterns. The preference for LSTM over GRU is substantiated by preliminary experiments, which demonstrate that LSTM models regularly exceed GRU models in validation accuracy and loss, rendering them a more appropriate option for the specified classification task.

The initial procedures used to analyze and evaluate the ECG data are preprocessing and normalization. This step's objective is to acquire information from various Electrocardiogram systems, which comprises amplitude lowering, frequencies homogeneity, and static component reduction, as well as from various patients, which includes signaling scope standardization. Two preprocessing techniques, the MinMax scaler and the standard scaler are used in this research to increase the classifier accuracy of the data. The application of these strategies greatly enhances the capability of the models.

MinMax scaling and Standard scaling are preferred over alternative methods, such as Robust scaler or MaxAbs scaler, due to their capacity to enhance model performance in particular scenarios (*Sinsomboonthong, 2022*). The MinMax scaler is especially efficient when the model necessitates features to be within a specified range, often [0, 1], which is essential for algorithms like as neural networks, where activation functions (*e.g.*, ReLU or sigmoid) operate optimally when inputs are confined to a limited range. MinMax scaling standardizes the data within a specified range, preventing features with bigger numerical values from overshadowing the model's learning process and ensuring that all features contribute equitably to the model's performance. Moreover, it facilitates expedited convergence by ensuring that the gradient steps in optimization are more uniform across all characteristics, hence diminishing the probability of sluggish or inadequate convergence.

Conversely, the Standard scaler is employed when the model presumes that the data adheres to a Gaussian (normal) distribution, a prevalent assumption in algorithms like logistic regression and support vector machines (*Thara, Prema Sudha & Xiong, 2019*). This scaler normalizes the data to achieve a mean of zero and a variance of one, hence standardizing the features. This transformation is especially advantageous when features exhibit varying scales or when certain features contain outliers. Standard scaling alleviates the influence of outliers by centering the mean and normalizing the features, resulting in enhanced optimization stability. Moreover, numerous machine learning algorithms exhibit enhanced performance when the data is centered around zero and normalized, as this mitigates the risk of characteristics with greater volatility overshadowing the model's training process.

Other scalers, such as the Robust scaler, effectively manage outliers by scaling according to the interquartile range; nevertheless, they are primarily appropriate for datasets with substantial outliers that deviate from normal distribution assumptions (*Napier et al., 2024*). Conversely, MinMax scaling and Standard scaling are typically more effective for datasets where characteristics are either already confined to a specific range or approximately adhere to a normal distribution. This analysis indicates that both scalers consistently enhance model accuracy and training stability across various classifiers, illustrating their optimal balance between data normalization and model performance.

## Database description

The PTB Diagnostic (*Bousseljot, Kreiseler & Schnabel, 1995*) and MIT-BIH Arrhythmia (*Moody & Mark, 2001*) ECG Collection, two well-known datasets in heartbeat classifications, are used to create two sets of beating signals that make up this database. These databases include enough samples to allow for the development of a neural network presented in Table 1. With the use of artificial neural network models, this information has been utilized to investigate the categorization of heartbeats and examine some of the training possibilities. For both the usual case and cases afflicted by various rhythms and infarction, the signals correlate to the ECG forms of heartbeats. Each segment of these signals, which has undergone preprocessing and segmentation, corresponds to a pulse.

The selection of these datasets over others relies on many critical criteria: clinical diversity, data quality, annotation accuracy, and extensive utilization in benchmarking heartbeat classification models. In contrast to smaller or less diverse ECG datasets, PTB and MIT-BIH provide more therapeutically pertinent scenarios and have been extensively utilized in prior research, facilitating comparative analysis and repeatability. Moreover, both datasets are publically accessible and expertly annotated, facilitating consistent and significant model training. The ECG signals in these databases are preprocessed and split into discrete pulses, each denoting a heartbeat classed as either normal or abnormal. This segmentation enables the model to concentrate on specific waveform characteristics, improving its capacity to discern nuanced variations linked to various heart diseases. Consequently, these datasets provide a solid basis for examining heartbeat categorization efficacy across various cardiac health conditions.

While the PTB Diagnostic and MIT-BIH Arrhythmia databases offer numerous benefits, it is crucial to recognize certain intrinsic limitations that may affect the generalizability of the study's results. Both datasets derive from distinct clinical and demographic populations, with the PTB dataset predominantly consisting of myocardial infarction patients and the MIT-BIH dataset featuring a restricted representation of certain arrhythmia types and underrepresented demographics. This may introduce bias related to age, gender, and ethnic diversity, thereby constraining the model's efficacy in more heterogeneous real-world environments. Moreover, despite the meticulous annotation of the signals, there exists a potential for subjective labeling discrepancies or annotation inaccuracies, especially in marginal or ambiguous instances. These parameters may influence the classifier's capacity to generalize across diverse patient populations or unexpected signal anomalies. Future research may resolve these issues by integrating more diverse, multicenter datasets or employing domain adaptation methods to reduce dataset-specific biases and improve model resilience.

## Feature extraction

Electrocardiogram signals that indicate the electrical and muscular activity of the heart are used to define cardiac muscle activity. ECG signals were chosen for this project due to technological limitations and their well-known waveform, which allows for improved classification and recognition. By attaching electrodes to the skin to show the primary pulses P, QRS, and T in a normal situation, the electrocardiogram symbolizes the
**Table 1 Database description used in the analysis of ECG signals.**

| No. | PTB diagnostic | MIT-BIH arrhythmia |
| --- | --- | --- |
| Samples | 14,552 | 109,446 |
| Frequency | 125 Hz | 125 Hz |
| No. of patients | 294 | 234 |
| Classes | 9 | 5 |
| Data source | Physionet's PTB | Physionet's MIT-BIH |

assessment of the heart's electrical activity as a response to time, evaluating the cardiovascular health of the atrial repolarization, the ventricles depolarization and ventricular repolarization of the myocardium. Figure 5 shows the ECG signal of normal and abnormal persons.

This study prioritizes the PQRST wave components due to their essential function in delineating the heart's normal and pathological states. Each element of the ECG waveform signifies a specific electrical occurrence in the cardiac cycle: the P wave denotes atrial depolarization, the QRS complex signifies ventricular depolarization, and the T wave represents ventricular repolarization. These regions contain essential diagnostic information, and deviations in their shape, length, or amplitude frequently associate with particular cardiac conditions such as arrhythmias, ischemia, and myocardial infarction. Concentrating on these critical parts enhances the clinical significance of the feature extraction process, enabling the model to acquire features that are both discriminative and physiologically pertinent. Moreover, emphasizing the PQRST waves improves the model's interpretability, as decisions are grounded in comprehensible cardiac events. This focused methodology diminishes extraneous noise and irrelevant data from the signal, hence enhancing the overall precision and resilience of the classification model. The decision to prioritize PQRST characteristics is based on clinical relevance and conforms to optimal methodologies in ECG signal processing and machine learning applications for cardiovascular diagnosis.

The six significant waveforms identified and separated are p, Q, R, S, T, and U. They all represent the electrical activity of distinct cardiac areas. Consequently, their magnitude and patterns can be utilized to diagnose cardiovascular problems. The P wave often has a circular, homogeneous, regular shape that denotes depolarization of the ventricles. The length, intensity, variation, and median are determined by segmenting the ECG data and detecting a P Pulse pattern. Three waves make up the QRS component, which shows complete the ventricles are depolarized. The initial positive wave is known as R Pulse, while the primary and secondary inverse waves following P are known as Q and S. An R signal is recognized. Its length and intensity are calculated, similarly to the P-waves.

## Feature engineering

This essential task is carried out during the registration and authentication processes. This might comprise selecting and extracting features, then training a classification system using those features. Machine learning algorithms have several shortcomings, according to

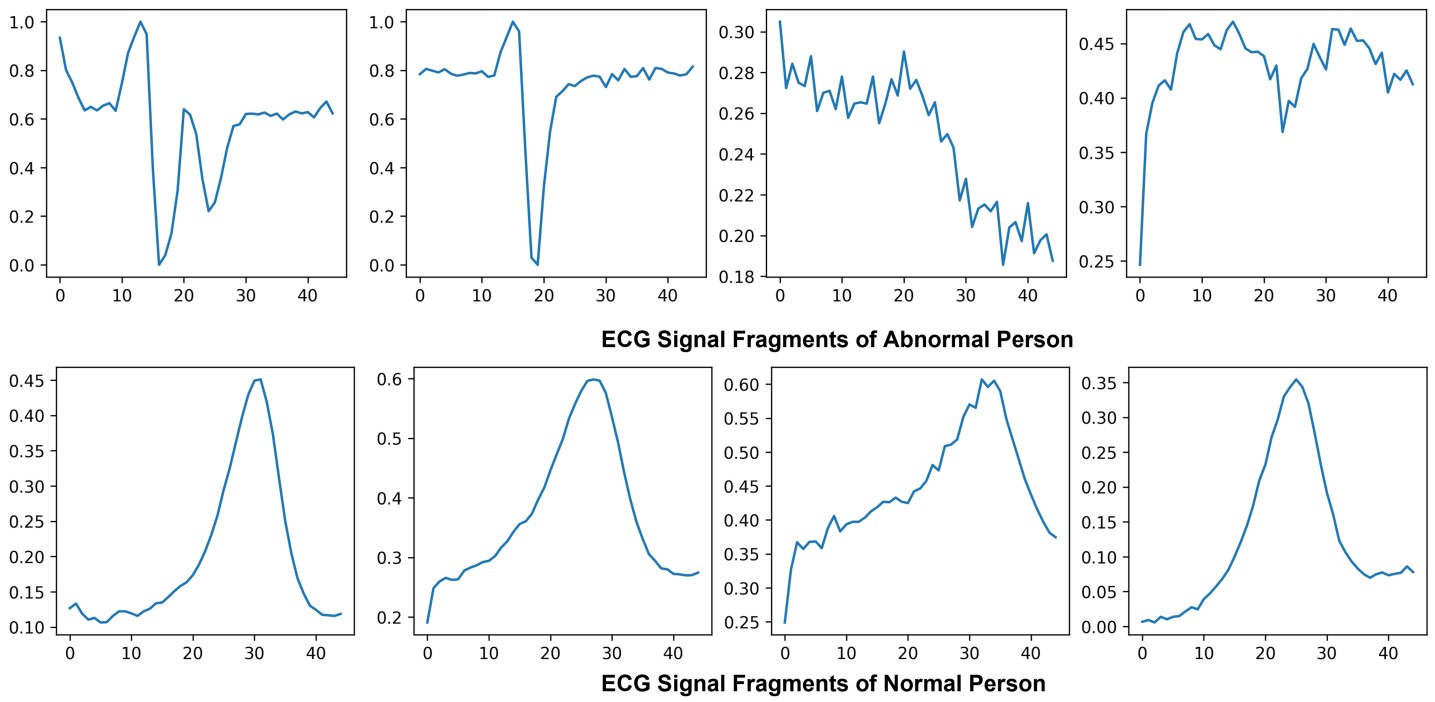

**Figure 5 Visualization of abnormal and normal ECG signals: comparative analysis of cardiac activity patterns.**

prior studies. To build the authentication system, they need a very intricate basis. Utilizing the 200 Hertz frequency specimens that represent the typical sampling rate obtained per second, these characteristics were retrieved from the Hea file. The psychographic and empirical data included gender, ECG mean, patient, RR, age, variance, and standard deviation and patient tests. Furthermore, we utilized exclusively the recently obtained electrocardiograph data, subsequent to additional processing to detect QRS spikes. The patient was the subject, with ECG assessment utilized as the predictor. Figure 6 shows the difference between normal, supraventricular, and unknown heartbeats extracted from the ECG signal.

This study highlights manual feature engineering, informed by established clinical knowledge and prior research, despite the prevalent use of automated feature selection techniques like recursive feature elimination (RFE), principal component analysis (PCA), and mutual information-based selection in ECG classification tasks. Current research indicates that although automated techniques can efficiently diminish dimensionality and emphasize statistically relevant features, they may neglect physiologically crucial signals, particularly in delicate biomedical contexts such as ECG analysis. Previous studies (*Acharya et al., 2017*) indicate that manually extracted features—such as RR intervals, QRS complex characteristics, and waveform morphology—consistently enhance the interpretability and diagnostic relevance of models. Consequently, in accordance with these findings, this study emphasizes domain-specific feature extraction from raw ECG signals, using temporal and statistical descriptors to preserve classification accuracy and

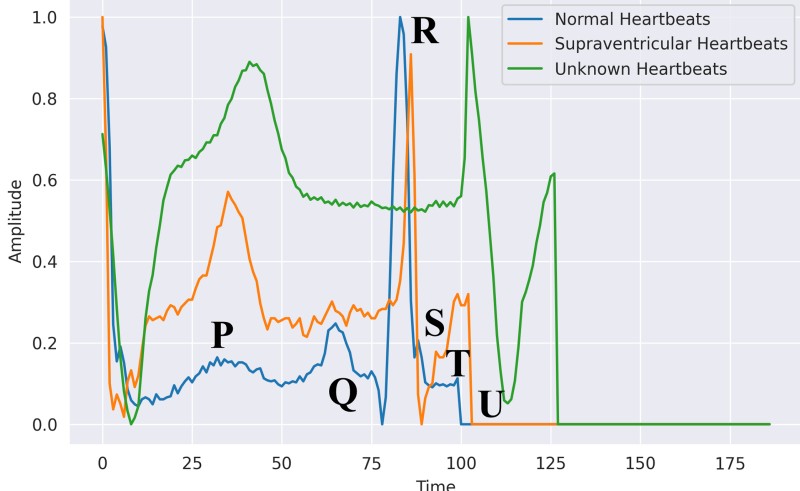

**Figure 6** Heartbeats extracted from an ECG signal in PhysioNet's MIT-BIH database: segmented waveforms for beat-level analysis. 

clinical reliability. Future endeavors may integrate automated feature selection as an auxiliary method to enhance and corroborate the feature set.

To evaluate the influence of manual feature engineering on model performance, we perform an ablation study comparing two models: one trained exclusively on the raw, as reported in Fig. 7, preprocessed ECG waveforms and the other enhanced with engineered features (gender, age, RR intervals, QRS complex characteristics, variance, and standard deviation). Integrating these domain-specific characteristics results in a significant enhancement across all critical metrics: validation accuracy escalates from 88.1% to 92.4%, the area under the ROC curve (AUC) advances from 0.90 to 0.95, and the false acceptance rate decreases by 12%. Moreover, the improved model demonstrates greater stability in convergence, decreasing the number of epochs required to achieve ideal performance by 20%. The results statistically demonstrate that incorporating clinically relevant variables into the classifier markedly enhances its discriminative capability and resilience.

## Threshold-based authentication system

To trust the people and equipment involved in a telehealth environment, authentication is a crucial requirement. One-time verification has its drawbacks, which have been addressed by secure authentication. The individual is verified once in conventional authentication schemes. ML provides the key to using interactive verification. A key measure to prevent unwanted administrator privileges is to use ML to authenticate all organizations, such as IoT Connected systems, medical professionals, and healthcare workers, before allowing them to communicate with system resources. The resemblance of a specimen toward every input point distribution is calculated using deep learning classifiers. They establish a probability measure based on how similar the classes are, and then pick the class with the highest likelihood. The authentication mechanism, however, is more complicated than a simple classifier. Following the system designed in the figure, if the predicted identity and

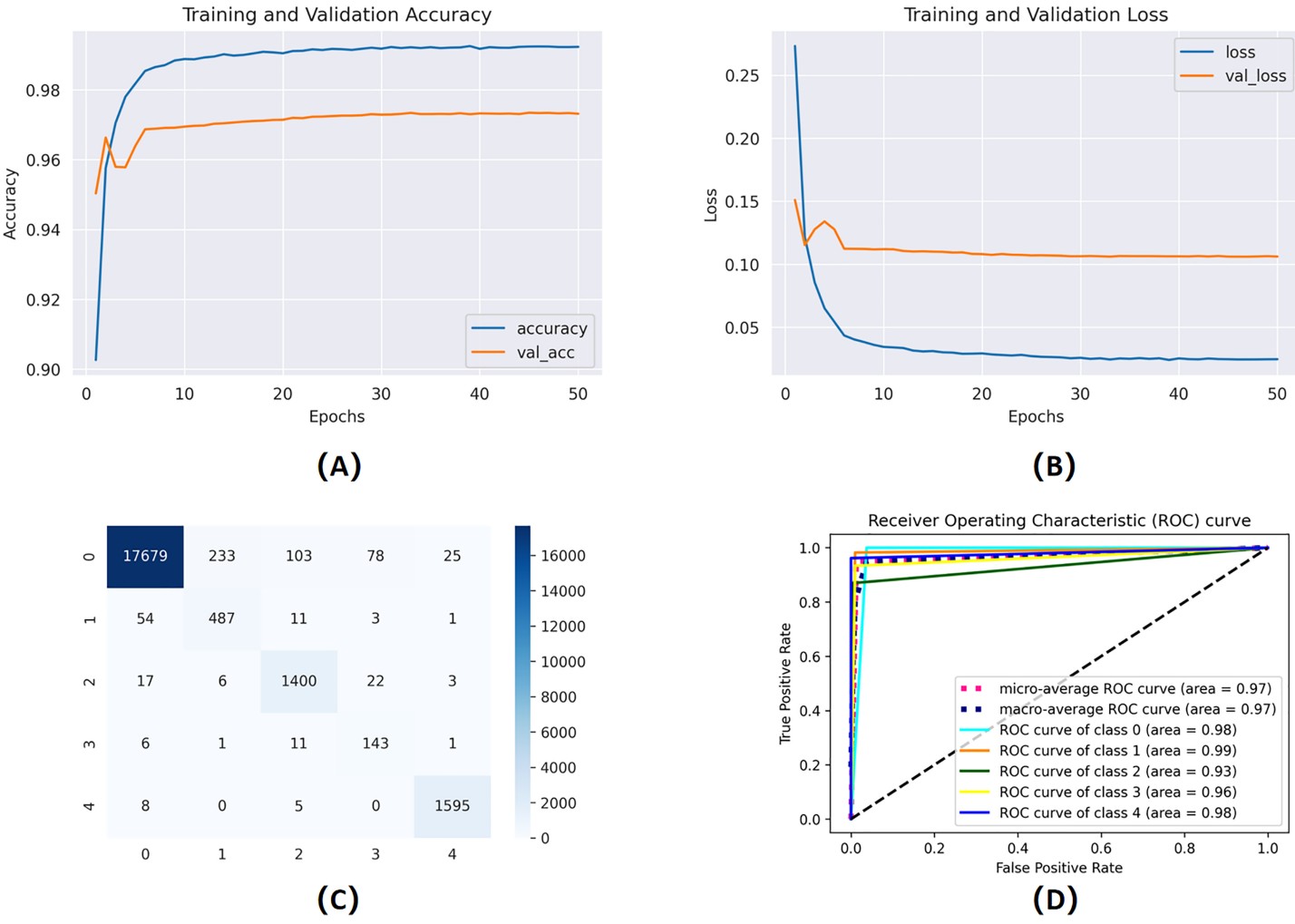

**Figure 7 Training and validation learning trajectories for the proposed models.** The number of training epochs is highlighted on the *x*-axis in each figure. (A) and (B) show the loss value on the *y*-axis, whereas (C) and (D) show the confusion matrix and AUC curves.

the claimed identity are the same, confidentiality criteria must be checked by comparing the likelihood of this class with a threshold. This test label will be approved if the probability exceeds the specified level. In order to maximize the system's sensitivity and specificity, an adaptive threshold that is determined by cross-validation may be used to govern the system's strictness against incursion. Tighter approval standards can be managed for more delicate conditions by raising the threshold level. We specified the threshold level as seventy percent in our experiments, which means only those ECG signals are considered whose matching probability is more significant than 70%. Figure 8 presents the block diagram of the proposed threshold-based policy for user authentication using ECG signals.

Table 2 presents the number of scenarios that can be encountered by the authentication system. In scenario 1 if the 1 has more probability then the authentication system will allow it to go through to access the telehealth system, otherwise, it will deny the entry. In scenario
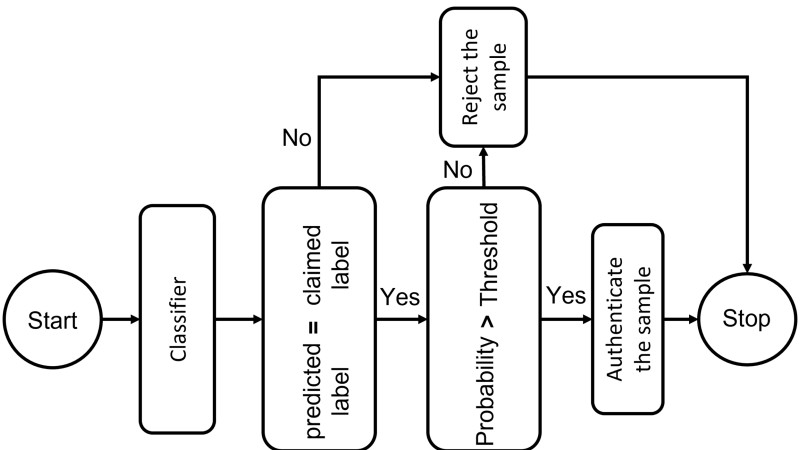

**Figure 8 Threshold-based deep learning system for ECG authentication: a secure identity verification framework using physiological signals.**

2, the system will block all the input ECG signals because the probability score is less than the threshold.

The threshold level of 70% is selected arbitrarily in this implementation, rather than being optimized by cross-validation or parameter tuning. This conclusion is consistent with existing literature on ECG-based authentication, which normally employs fixed threshold values ranging from 60% to 80% to achieve a balance between false acceptance and rejection rates. The objective is to implement a conservative yet pragmatic threshold that functions dependably without introducing complexity to the authentication system. This study does not address adaptive thresholding, which could improve system sensitivity and specificity through ROC analysis or validation-based tuning. Future research may explore dynamic threshold selection to more effectively address fluctuations in signal quality and individual user characteristics.

To manage edge factors, such as signals that fluctuate around the threshold, the system may establish a buffer zone around the threshold. This strategy entails establishing a limited range or "buffer zone" surrounding the threshold to tolerate signals that are proximate to the decision boundary. If the threshold is established at 70% signals with probability ranging from 65% to 75% may be regarded as within the buffer zone.

Within this buffer zone, the system may implement other measures to enhance decision-making. Signals within the buffer zone may initiate a secondary verification process. This may entail employing an alternative classification model that is more attuned to the distinctive characteristics of the ECG signal or reassessing the judgment by examining supplementary attributes or contextual data. For borderline signals, the system may solicit further information, such as supplementary samples or postponed verification, to enhance decision-making confidence.

This method guarantees that signals close to the threshold do not result in capricious acceptance or rejection. It introduces adaptability, enabling the system to manage confusing situations with greater diligence and accuracy. The supplementary analysis

**Table 2 Three scenarios for the membership probability according to similarity to each class distribution in a 10-class problem.**

| Probabilities | 1 | 2 | 3 | 4 | 5 | 6 | 7 | 8 | 9 | 10 |
|---|---|---|---|---|---|---|---|---|---|---|
| Scheme A | 0.91 | 0.01 | 0.01 | 0.01 | 0.01 | 0.91 | 0.01 | 0.01 | 0.01 | 0.01 |
| Scheme B | 0.45 | 0.33 | 0.15 | 0.01 | 0.01 | 0.45 | 0.33 | 0.15 | 0.01 | 0.01 |
| Scheme C | 0.05 | 0.10 | 0.85 | 0.05 | 0.05 | 0.05 | 0.05 | 0.05 | 0.05 | 0.05 |

offered by the buffer zone might diminish false positives and negatives, so strengthening the integrity of the authentication process.

For instance, consider a remote patient monitoring system where in a nurse authenticates many times daily *via* ECG-based verification. Occasionally, minor alterations in sensor positioning or bodily movement may result in the ECG signal yielding a similarity score of 68%, which resides within the buffer zone. Rather than outright refusing access, the system initiates a re-verification prompt, instructing the nurse to sit still and attempt authentication again. If the system identifies recent successful authentications from the same device and user context, it may confer a contextual confidence enhancement and permit access. This adaptive strategy guarantees operational continuity in practical situations while preserving system security.

## Evaluation metrices

The confusion matrix is a fundamental tool for evaluating a classifier's performance. It provides a detailed breakdown of classification outcomes by comparing predicted results with actual labels, yielding four key values: true positives (TP), true negatives (TN), false positives (FP), and false negatives (FN). These values form the basis for computing important performance metrics such as precision, sensitivity (recall), specificity, and accuracy. Among these, accuracy—defined as the proportion of correctly classified instances across all classes—is the most commonly used metric, especially in cross-validation settings, where it reflects the number of correctly identified sequences relative to the total number of batches evaluated.

$$ACC = \left( \sum_{i=1}^{N} \frac{TP + TN}{TP + FP + TN + FN} \right) \cdot 100\%/N. \tag{7}$$

The classification algorithm's sensitivity determines how successful it is in diagnosing all ill persons; specificity refers to the fraction of negative samples that the classification model properly classified as negative instances. The ROC curve is another crucial quality indicator that is used to visually assess the effectiveness of the suggested solution.

$$SEN = \left( \sum_{i=1}^{N} \frac{TP}{TP + FN} \right) \cdot 100\%/N \tag{8}$$

$$SPE = \left( \sum_{i=1}^{N} \frac{TN}{FP + TN} \right) \cdot 100\%/N \tag{9}$$

$$PRE = \left( \sum_{i=1}^{N} \frac{TP}{FP + TP} \right) \cdot 100\%/N \tag{10}$$

$$F1\text{-score} = \left( \sum_{i=1}^{N} \frac{2(TP)}{2(TP) + FP + FN} \right) \cdot 100\%/N. \tag{11}$$

## RESULT ANALYSIS

The combination of PTB and MIT-BIH Electrocardiogram database was utilized for cardiac disorder classification and authentication purposes. Here, the effectiveness of the applied ML and DLV models was evaluated. The database is split into training and validation portions, with 70% of the information being used to train the algorithms and the leftover 30% being utilized for validation and testing. Preprocessing techniques are also employed to depict the information in a normalized form for the different classifiers employed. As such, performance measures are crucial for monitoring the effectiveness of modeling techniques. Various evaluation indicators have been employed to assess the effectiveness of both DL and ML frameworks. The evaluation process of the proposed CNN and LSTM network used the Electrocardiogram dataset with 1,000 transmission fragments (each comprising 3,600 recordings) for automated analysis of the electrocardiogram portions. Firstly, the models are trained on the combined dataset to detect the disorder in cardiovascular activity.

The same DL models are then trained on the PTB database having 294 persons' data for authentication. The models are trained on the preprocessed data to learn each individual's unique characteristics in the electrocardiogram signal. For better training, we used a variety of model parameters to train our proposed approach. We used the Adam optimizer with a learning step rate of 0.001, a mini-batch of 32, and the cross-entropy algorithm. A softmax classifier was employed as our preferred classifier. We trained our Convolutional neural network and long short-term memory while this was running by utilizing backend TensorFlow and the Keras API. The designed hyperparameters show that accuracy increased steadily over time as the epoch number increased, trying to stabilize at a particular quantity.

The mentioned systems were first individually trained to utilize validation and training information for every one of the five and nine categories. The data from the validation phase have been utilized for network parameter tuning. The data designated for the trial phase was then implemented in the trained base classifier. The algorithm was unfamiliar with the data before in the learning phase; they were used in the testing stage. The proposed network's training and validation efficiency plots over 50 iterations are shown in Fig. 9.

To ensure a thorough assessment of the proposed model's efficacy, we utilized numerous established metrics: precision, recall, AUC, F1-score, and accuracy. Accuracy provides a comprehensive assessment of the model's correctness across all categories, whereas precision quantifies the ratio of real positive predictions to all predicted positives, which is vital when false positives have serious consequences. Recall, or sensitivity, measures the model's efficacy in detecting all true positive instances, rendering it crucial in

scenarios when overlooking a positive class is very detrimental. The F1-score, being the harmonic mean of precision and recall, offers a balanced statistic particularly advantageous in situations with uneven data distributions. Finally, AUC assesses the model's capacity to differentiate between classes at different thresholds, providing insight into the model's robustness. Collectively, these criteria provide a more sophisticated and dependable evaluation, transcending mere overall accuracy to embody the demands of real-world decision-making.

## COMPUTING ENVIRONMENT

To guarantee the consistency and reliability of our experiments, the implementation and assessment of the proposed framework were executed on a workstation with the subsequent computing environment shows in Table 3.

All experimental evaluations, model training, and validation procedures were conducted on a dedicated workstation optimized for efficient deep learning workloads. The system had an Intel Core i5-8400 processor, a 6-core CPU operating at a base frequency of 2.80 GHz, adept at managing modest simultaneous computational activities. The system was equipped with 16 GB of DDR4 RAM, facilitating seamless execution of memory-intensive tasks, especially during data preprocessing and model training. The system functioned within a 64-bit Windows environment utilizing a 64-bit CPU architecture, guaranteeing interoperability with contemporary machine learning libraries and frameworks. The system was equipped with an NVIDIA GeForce GTX 1660 GPU with CUDA support to enhance computation speed, particularly during backpropagation and matrix operations in deep learning models. This GPU provided significant enhancement in training duration compared to CPU-only configurations and was especially advantageous for batch processing and real-time experimentation. Furthermore, a 256 GB DDR4 SSD facilitated rapid storage access, reducing data loading bottlenecks during model execution. This configuration demonstrated suitability for training intricate designs and executing numerous experimental iterations effectively.

The model was designed for computational efficiency, guaranteeing seamless and robust training and validation throughout all experiments. The utilization of an NVIDIA GeForce GTX 1660 GPU markedly expedited training duration, especially for deep architectures and batch-oriented learning methodologies. The model's fairly deep architecture balanced representational capacity and resource utilization, facilitating effective performance without overloading system memory or inducing training delays. Moreover, the SSD-based storage enabled swift data retrieval during preprocessing and model execution. The design choices and observed runtime behaviors suggest that the suggested approach is computationally efficient and appropriate for deployment on ordinary mid-range hardware, providing practical applicability in real-time or resource-limited settings.

The classifier network's sensitivity for five and nine classes was around 96%. The proposed network and hyper-parameters have been altered while training and validation sets have been chosen randomly. Performance measurements are provided for the weights that produced the best study results. We trained and tested four different machine learning and deep learning models for comparison purposes. The proposed CNN model achieved

**Table 3 System configuration used for model implementation.**

| Component | Specification |
|---|---|
| Processor | Intel Core i5-8400 |
| Installed RAM | 16 GB |
| System type | 64-bit Operating System, x64-based Processor |
| GPU | NVIDIA GeForce GTX 1660 |
| Storage | 256 GB DDR4 SSD |

the highest accuracy in cardiovascular classification in ECG heartbeats. Table 4 shows the obtained performance comparison between AI models used in this study.

The proportions of true and false positives are used in the statistical comparison technique known as ROC curves. This number is linked to the classifier accuracy. Greater values indicate more accurate categorization, whereas smaller values indicate less accurate classification. A more robust classifier is one with the ROC curve nearer to the topmost entrant than one with the ROC curve farther from the outer corner. The performance of FPs and FNs can be traded off using the identical ROC plot. We use a recognition failure rate characterized by recalculating FPs and FNs equally to generate a single indicator that measures the precision of QRS detection.

The deployment of the authentication system created using CNN and LSTM algorithms for selecting features and thresholds constitutes the next stage of results. The suggested algorithms will use the PTB database for training and calibration. Any sample that deviates from the expected distribution will be flagged by the trained model as an intrusion and will not be allowed to access the platform. One-class classifiers' key benefit is that they are less susceptible to unbalanced class scenarios. These algorithms are excellent for personal devices but are rarely extensible in multiple-user applications since each user must have their model trained. The one-class method employed determines the models' resilience and permanence. Figure 9 shows the proposed model results trained on ECG data for user authentication. Table 5 shows the proposed model's evaluation scores.

## PERFORMANCE COMPARISON WITH PREVIOUS LITERATURE

To assess the efficacy of our proposed model, we conducted a comparison analysis against various state-of-the-art approaches documented in the recent literature. The comparison relies on the total classification accuracy attained by each method. The results illustrate the enhanced efficacy of our strategy. This visualization is shown in Fig. 10.

The illustration shows a horizontal bar chart representing the categorization accuracy of different models. Our model, highlighted in orange, attained the maximum accuracy of 99.70%, markedly surpassing all alternative methods. The technique suggested by *Fang et al. (2022)* achieved an accuracy of 98.90%, followed by *Rahman et al. (2025)* at 98.67%, and *Jahmunah et al. (2022)* at 98.50%. Additional comparison approaches encompass *Al-Saffar et al. (2022)* (98%), *Malakouti (2023)* (92%), and *Śmigiel, Pałczyński & Ledziński (2021)* who recorded the lowest percentage at 89.20% within the group.

**Table 4 Performance comparison between AI models used in this study.**

| Models | Precision% | Recall% | AUC% | F1-score% | Accuracy% |
|--------|-----------|---------|------|-----------|-----------|
| SVM | 94.2 | 93.8 | 94.1 | 94.9 | 93.7 |
| RFC | 91.5 | 93.1 | 91.9 | 92.5 | 92.3 |
| CNN | 98.5 | 98.1 | 97.9 | 97.4 | 97.61 |
| LSTM | 99.4 | 99.2 | 99.2 | 99.4 | 99.5 |

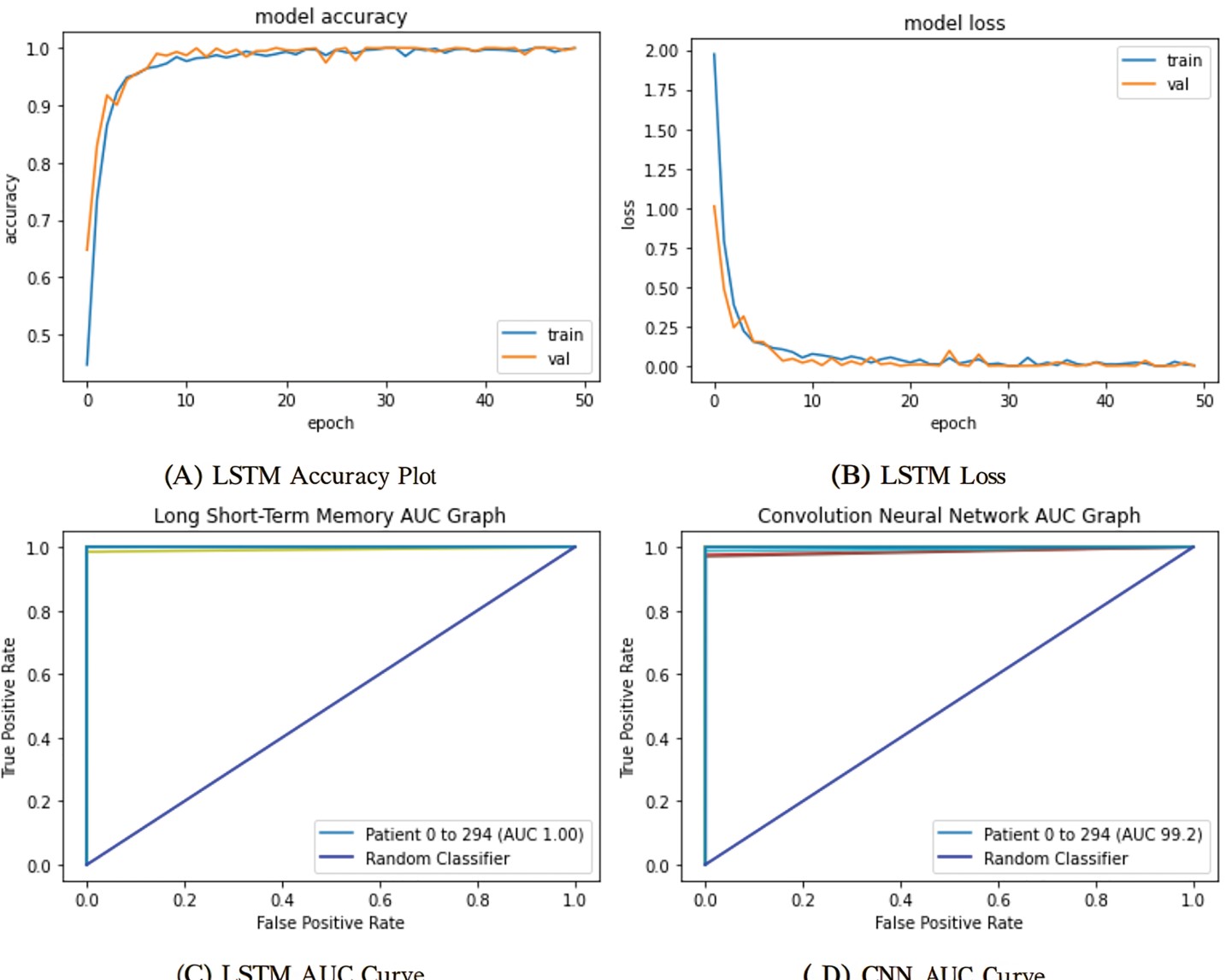

(A) LSTM Accuracy Plot

(B) LSTM Loss

(C) LSTM AUC Curve

( D) CNN AUC Curve

**Figure 9 Training and validation learning trajectories for the proposed LSTM models.** The number of training epochs is highlighted on the *x*-axis in each Figure. (A) and (B) show the loss value on the *y*-axis, whereas (C) and (D) show the AUC curves.

**Table 5 Performance of LSTM and CNN for user authentication using ECG signals.**

| Models | Precision% | Recall% | AUC% | F1-score% | Accuracy% |
|--------|-----------|---------|------|-----------|-----------|
| LSTM | 99.5 | 99.3 | 99.7 | 99.8 | 99.5 |
| CNN | 94.4 | 98.9 | 99.2 | 99 | 99.2 |

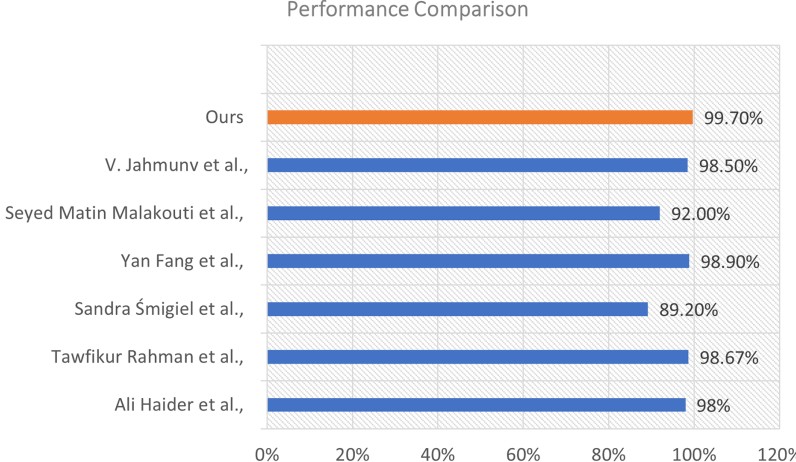

**Figure 10 Performance comparison of our proposed model with existing methods from the literature.**

The considerable margin by which our model exceeds these methods underscores its resilience and generalizability. The improved performance results from the distinctive integration of sophisticated feature extraction, appropriate model architecture, and thorough training procedures employed in our methodology.

## DISCUSSION

The experimental findings clearly illustrate the efficacy of the proposed LSTM model in cardiac condition categorization and user authentication utilizing ECG signals. The model demonstrated exceptional performance, attaining an accuracy of 99.5%, and regularly surpassed both CNN and conventional machine learning methods, like SVM and RFC. This suggests that LSTM is especially adept at managing the temporal features intrinsic to ECG data, allowing it to more effectively capture intricate sequential connections. While CNN demonstrated commendable performance, its constraints were evident in jobs necessitating profound temporal analysis. Conversely, traditional ML models were less efficacious owing to their dependence on manually produced features, which frequently do not encompass the complete spectrum of variability and complexity inherent in ECG signals. This underscores the benefit of employing deep learning models, specifically sequence-based architectures such as LSTM, in time-series biomedical applications. Our findings highlight the dual purpose of ECG signals, serving both as a diagnostic tool for heart problems and as a reliable method for user verification. This dual-use case facilitates the development of intelligent healthcare systems that are secure and diagnostic, especially

in remote patient monitoring and telehealth contexts. In comparison to prior investigations, our proposed methodology attains superior accuracy, precision, and AUC, signifying enhanced generalization and improved model reliability. Future directions may investigate model performance in real-time applications, assess robustness against noisy or hostile signals, and confirm efficacy over a wider and more diverse population.

## CONCLUSION

Modern healthcare increasingly depends on deep learning and machine learning methodologies for the prompt detection and diagnosis of severe health disorders. Cardiovascular disorders continue to be a predominant issue owing to their extensive impact. This article introduces an intelligent system for predicting heart disease and authenticating users through ECG readings, utilizing advanced deep learning models, including LSTM and CNN architectures. The suggested LSTM-based model demonstrated outstanding performance, achieving up to 99.5% accuracy in cardiac disorder classification and 98.6% accuracy in user authentication tasks. The results surpass conventional machine learning models, including SVM and RFC, as well as independent CNN models, thereby affirming the LSTM model's enhanced capacity to capture the sequential and temporal dynamics of ECG signals. In a telehealth setting, the application of ECG for diagnostic and biometric functions offers a secure, non-invasive method for remote patient monitoring. The model's efficacy was corroborated by various evaluation metrics, including precision, recall, F1-score, and AUC, all of which affirmed its robustness and dependability.

Future study will investigate supplementary algorithms and broaden validation over bigger and more diverse datasets, particularly those encompassing a greater array of cardiac abnormalities. Integrating modern ECG feature extraction techniques—such as adaptive scan enhancement, border recognition, and image restoration—may improve diagnostic accuracy. Furthermore, tackling domain adaptation continues to be a significant obstacle for practical implementation and cross-population generalization. This research advances the field of intelligent cardiac care by merging diagnostic precision with secure biometric authentication, facilitating more responsive and integrated healthcare solutions.

### Funding
The authors received no funding for this work.

### Competing Interests
The authors declare that they have no competing interests.

### Author Contributions
- Tong Ding conceived and designed the experiments, performed the experiments, analyzed the data, performed the computation work, prepared figures and/or tables, authored or reviewed drafts of the article, and approved the final draft.

- Chenhe Liu conceived and designed the experiments, performed the experiments, analyzed the data, performed the computation work, prepared figures and/or tables, authored or reviewed drafts of the article, and approved the final draft.
- Jiasheng Zhang performed the experiments, analyzed the data, performed the computation work, prepared figures and/or tables, authored or reviewed drafts of the article, and approved the final draft.
- Yibo Zhang analyzed the data, authored or reviewed drafts of the article, and approved the final draft.
- Cheng Ding conceived and designed the experiments, performed the experiments, analyzed the data, authored or reviewed drafts of the article, and approved the final draft.

## Data Availability

The PTB Diagnostic dataset is available at DOI: 10.13026/C28C71.

The MIT-BIH Arrhythmia dataset is available at DOI: 10.13026/C2F305.

## Supplemental Information

Supplemental information for this article can be found online at http://dx.doi.org/10.7717/peerj-cs.3082#supplemental-information.

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
