# Peer review of "Deep learning based cardiac disorder classification and user authentication for smart healthcare system using ECG signals"

_PeerJ Computer Science, doi:10.7717/peerj-cs.3082_

## Round 0.1 · original submission · Major Revisions

Dear Authors,

Thank you for submitting your article. Reviewers have now commented on your article and suggest major revisions. We do encourage you to address the concerns and criticisms of the reviewers and resubmit your article once you have updated it accordingly.

Best wishes,

Reviewer 1 ·

Basic reporting

- Consider rephrasing for conciseness and clarity. For instance: "Abnormal cardiac activity can lead to severe health complications, emphasizing the importance of timely diagnosis."
- Replace "promises to revolutionize" with more objective language, such as "has the potential to transform," to maintain a professional tone.
- "Abnormal cardiac activity is a potentially fatal condition that leaves individuals with serious health issues." This sentence is repeated verbatim in the abstract and introduction. Avoid redundancy to improve the paper's flow.
- Consider citing additional recent references to support the widespread adoption of AI in these fields.
- Elaborate briefly on these "several characteristics" to provide a clearer link between prior research and your contribution.
- Ensure all references are recent, relevant, and formatted according to journal guidelines.
- Some descriptions are verbose and could benefit from concise language for improved readability.

Experimental design

- The methods are described in enough detail to allow reproducibility. However, some critical parameter choices, such as why specific layer configurations or threshold levels were selected, could use further justification.
- The description of parameter tuning and layer adjustments for achieving optimal accuracy is too general. Consider elaborating on how these adjustments were made, what criteria were used, and whether grid search, random search, or another optimization strategy was applied.
- The CNN architecture is clearly outlined, but the rationale behind choosing specific configurations (e.g., filter size, dropout rates, and activation functions) is missing.
- It is unclear whether the 5 output units correspond to specific classes. Clarify how the softmax layer outputs were interpreted.
- The explanation of LSTM layers and their configurations is robust but could benefit from a discussion of why LSTMs were chosen over other recurrent models like GRUs.
- The use of exponential terms in LSTM (Line 213) should be further elaborated—what specific advantage does this bring to ECG signal processing?
- The use of MinMax and standard scalers is briefly mentioned. It would be helpful to clarify why these specific scaling methods were selected over others and how they impact model performance.
- Were any data augmentation techniques employed during preprocessing to address potential class imbalances?
- The use of PTB Diagnostic and MIT-BIH datasets is appropriate, but the rationale for selecting these datasets over others should be discussed.
- The features extracted for classification are well-described, but more emphasis is needed on why certain features (e.g., PQRST waves) were prioritized.
- Were automated feature selection methods evaluated? If so, how did they compare to manual feature engineering?
- The threshold level of 70% is mentioned as an experimental choice. Explain whether this was optimized through cross-validation or chosen arbitrarily.
- How does the system handle edge cases, such as signals that hover around the threshold? Are there fallback mechanisms?
- The metrics described are appropriate, but consider including additional metrics such as ROC-AUC, or Matthews Correlation Coefficient to provide a more comprehensive evaluation.

Validity of the findings

- While the CNN and LSTM architectures are well-detailed, there is little discussion on why specific configurations (e.g., number of layers, dropout rates, filters) were chosen.
- Terms like "ECG identification," "authentication," and "classification" are used interchangeably, which may confuse readers.
- While metrics like accuracy, precision, sensitivity, and specificity are mentioned, no quantitative results are provided in this section.
- The description of the threshold-based authentication system lacks detail regarding why a 70% threshold was chosen and how adaptive thresholds were implemented.
- The formulas for LSTM (line 214) lack adequate explanation or context.
- While feature engineering is mentioned, its quantitative impact on model performance is not discussed.

Additional comments

- While the content is detailed and provides a comprehensive explanation of the methods and models used, some sections are overly verbose.
- Terms such as "1-dimensional CNN," "Cnn," and "Cnn model" are used inconsistently. Standardizing the terminology throughout the manuscript would improve clarity and professionalism.
- The manuscript contains several grammatical errors and awkward sentence constructions. A thorough proofreading is recommended to address issues like subject-verb agreement, word choice, and sentence structure. For example, "The latest recent developments in AI" could simply be "The latest developments in AI."
- The explanation of the threshold-based authentication system is clear, but additional examples or real-world scenarios could help readers understand its practical applications and limitations.
- While the datasets used are described, consider including a brief discussion of any potential biases or limitations in these datasets and how they might impact the study's results.

Reviewer 2 ·

Basic reporting

The paper does not compare the proposed method's performance with other existing ECG classification techniques.

Adding a comparison table with benchmark models would provide a better perspective on improvements.

Experimental design

While the paper mentions changes in layers and hyperparameters, it does not justify why specific values (e.g., 64 filters, 6 map size) were chosen.

Including an ablation study or results from different hyperparameter settings would strengthen the analysis.
Were additional metrics such as F1-score, AUC-ROC, or mean squared error considered? Justifying the choice of evaluation metrics based on the problem type would strengthen the discussion.

Validity of the findings

All underlying data have been provided; they are robust, statistically sound, & controlled.

The data on which the conclusions are based must be provided or made available in an acceptable discipline-specific repository. The data should be robust, statistically sound, and controlled.
Conclusions are well stated, linked to original research question & limited to supporting results.

Additional comments

While accuracy is important, was there an evaluation of computational efficiency? Discussing time complexity and memory usage could add value to the practical implications of the method.

Reviewer 3 ·

Basic reporting

The manuscript explores deep learning for cardiac disorder detection and secure user authentication using ECG signals. It provides a clear introduction, strong literature support, and well-structured content. The writing is precise, with high-quality figures and proper references.

Experimental design

The manuscript uses the PTB Diagnostic and MIT-BIH Arrhythmia datasets to train and evaluate deep learning models for cardiac disorder classification and user authentication. The authors employ CNN and LSTM architectures, incorporating data preprocessing and augmentation to enhance model performance. Model assessment is conducted using accuracy, precision, recall, F1-score, and AUC metrics, ensuring a comprehensive evaluation.

Validity of the findings

The manuscript validates its findings using PTB Diagnostic and MIT-BIH Arrhythmia datasets for both classification and authentication. CNN and LSTM models achieve high accuracy, with the authentication system using a threshold-based classifier for reliable ECG-based verification. Strong performance metrics and thorough evaluation confirm the model’s effectiveness.

·

Basic reporting

The manuscript follows the journal's criteria and is well-structured. It focuses on deep learning models that classify heart diseases and authenticate users using ECG data. The comprehensive literature evaluation draws from several pertinent, peer-reviewed sources. Technically speaking, the study does a good job of describing CNN and LSTM models.

Nevertheless, there are a few things that could be done better. For example, certain grammar and linguistic problems need to be fixed. Additionally, certain sections may need more clarification, and some figures and diagrams are not very clear. More external research references might counterbalance the authors' heavy reliance on self-citations.

The authors should consider cutting out unnecessary words and increasing the general flow of the content to make it easier to read. Along with more thorough explanations of feature extraction in Section 3, including comprehensive captions for the figures would also be beneficial. Finally, adding more citations from other innovative studies would strengthen the paper.

The paper provides insightful information about deep learning models based on ECGs and is organized effectively.

Experimental design

The paper presents a deep learning-based model for user authentication and cardiac disease detection using publicly accessible datasets such as PTB and MIT-BIH. The authors describe the deep learning architecture, particularly the CNN and LSTM models, well and explain their decisions about model selection, threshold settings, and data augmentation methods.

However, there is room for improvement in a few areas. For example, the rationale behind the model and threshold selection is insufficient. A sensitivity analysis would be helpful to find out how alternative threshold values impact the model's performance. Furthermore, although the authors discuss several strategies for feature scaling, noise filtering, and ECG beat segmentation, they don't discuss them. Transparency would be improved, and these problems might be resolved by including a detailed representation of the preparation chain.

The utilization of publicly accessible datasets and the authors' apparent justification for their choices are the study's strong points, on the plus side.

Validity of the findings

With a classification accuracy of 99.5%, the suggested model outperforms several cutting-edge techniques. It also offers robust performance indicators supporting its efficacy, such as AUC, F1-score, sensitivity, and specificity. Interestingly, the model presents a novel method for user identification based on ECG.

Nonetheless, a few important issues must be resolved. Overfitting problems and the model's computational expense must be considered. The authors should address regularization techniques and provide graphs comparing training and validation loss to support their conclusions. Testing the model on external datasets would be helpful to further validate its performance.

Given that CNN and LSTM models can be highly resource-intensive, the authors should also discuss the computing time and hardware requirements. Another important consideration is the model's lack of clinical validation—it hasn't been tried in actual hospital settings or assessed for efficacy across various patient demographics.

Reviewer 5 ·

Basic reporting

1. The dataset features need to be described in more detail, including its total size and train/test split, preferably in a table. Additionally, the dataset should be made publicly accessible.
2. The manuscript requires a thorough proofreading to eliminate grammatical errors and typos.
3. The clarity and resolution of figures should be enhanced to improve readability and comprehension.
4. The authors should update the Related Work section with recent and relevant articles.

Experimental design

1. Pseudocode and algorithm steps should be included to clarify the methodology.
2. The architecture of the proposed model must be explicitly detailed.
3. The parameters used for the analysis should be summarized in a table.
4. The cost associated with deploying these deep learning models, including the required hardware and software, should be addressed.

Validity of the findings

1. The manuscript should report all relevant metrics in the experimental results for a comprehensive evaluation.
2. Time spent on training and testing should be measured and presented.
3. The accuracy and improvement percentages should be explicitly stated in the abstract and conclusion, along with an analysis of the significance of these results.
4. The limitations of the study should be discussed, along with potential improvements or challenges.

Additional comments

1. A discussion section should be included to provide a deeper interpretation of the results.
2. A future work section should be added at the end of the conclusion to highlight possible extensions of the research.

---

## Round 0.2 · accepted · Accept

Dear Authors,

Thank you for addressing the reviewers' comments. Your manuscript now seems sufficiently improved and ready for publication.

Best wishes,

·

Basic reporting

The manuscript was accurately revised, and it is clearly improved with the previous revision.

Experimental design

The methods are clearly described.

Validity of the findings

The findings are clearly validated with other studies.

Reviewer 5 ·

Basic reporting

Accept.

Experimental design

-

Validity of the findings

-

Additional comments

-